# DNA methylation repels targeting of *Arabidopsis* REF6

Qi Qiu [1,2,6], Hailiang Mei[1,6], Xian Deng[1,6], Kaixuan He[1,2,6], Baixing Wu[3,6], Qingqing Yao[3], Jixiang Zhang[2,4], Falong Lu [2,4], Jinbiao Ma [3] & Xiaofeng Cao [1,2,5]

RELATIVE OF EARLY FLOWERING 6 (REF6/JMJ12), a Jumonji C (JmjC)-domain-containing H3K27me3 histone demethylase, finds its target loci in *Arabidopsis* genome by directly recognizing the CTCTGYTY motif via its zinc-finger (ZnF) domains. REF6 tends to bind motifs located in active chromatin states that are depleted for heterochromatic modifications. However, the underlying mechanism remains unknown. Here, we show that REF6 preferentially bind to hypo-methylated CTCTGYTY motifs in vivo, and that CHG methylation decreases REF6 DNA binding affinity in vitro. In addition, crystal structures of ZnF-clusters in complex with DNA oligonucleotides reveal that 5-methylcytosine is unfavorable for REF6 binding. In *drm1 drm2 cmt2 cmt3* (*ddcc*) quadruple mutants, in which non-CG methylation is significantly reduced, REF6 can ectopically bind a small number of new target loci, most of which are located in or neighbored with short TEs in euchromatic regions. Collectively, our findings reveal that DNA methylation, likely acting in combination with other epigenetic modifications, may partially explain why REF6 binding is depleted in heterochromatic loci.

[1] State Key Laboratory of Plant Genomics and National Center for Plant Gene Research, Institute of Genetics and Developmental Biology, Chinese Academy of Sciences, Beijing 100101, China. [2] University of Chinese Academy of Sciences, Beijing 100049, China. [3] State Key Laboratory of Genetic Engineering, Collaborative Innovation Centre of Genetics and Development, Department of Biochemistry, Institute of Plant Biology, School of Life Sciences, Fudan University, Shanghai 200433, China. [4] State Key Laboratory of Molecular Developmental Biology, Institute of Genetics and Developmental Biology, Chinese Academy of Sciences, Beijing 100101, China. [5] Center for Excellence in Molecular Plant Sciences, Chinese Academy of Sciences, Beijing 100101, China. [6] These authors contributed equally: Qi Qiu, Hailiang Mei, Xian Deng, Kaixuan He, Baixing Wu. Correspondence and requests for materials should be addressed to J. M. (email: majb@fudan.edu.cn) or to X.C. (email: xfcao@genetics.ac.cn)

Polycomb-mediated trimethylation of histone H3 lysine 27 (H3K27me3), a conserved epigenetic mark associated with chromatin compaction and gene repression, plays a key role in cell identity and developmental regulation in multicellular eukaryotes[1–3]. Dynamic regulation of H3K27me3 at specific targets, which is essential for normal development, is achieved by balancing the activity of histone methyltransferases and demethylases of H3K27me3[4–6]. The RELATIVE OF EARLY FLOWERING 6 protein (REF6/JMJ12), a Jumonji C (JmjC) domain—containing histone demethylase, specifically demethylates H3K27me3 at its target loci. REF6 has intrinsic DNA-binding ability and specifically recognizes its target sequence (CTCTGYTY motif, Y = T or C) via tandem zinc-finger (ZnF) domains located at its C-terminus[7,8]. Target recognition by REF6 is required for recruitment of the SWI/SNF-type chromatin remodeler BRAHMA to the enzymes' common target loci[9]. However, CTCTGYTY motifs are not sufficient for REF6 recruitment, and only ~15% of such sequences in the *Arabidopsis* genome are bound by the enzyme[7], suggesting that an additional layer of regulation is involved in targeting REF6 in order to precisely control the level of H3K27me3 at developmentally important loci.

5-methylcytosine (5mC), is an evolutionarily conserved epigenetic mark. Accordingly, 5mC has long been considered the 'fifth base' in eukaryotic genomes, providing another layer of genome regulation[10]. DNA methylation fine-tunes gene expression and transposon silencing, playing important roles in maintenance of the structure and function of heterochromatin, genome stability, genomic imprinting, transgene silencing, and gene evolution[11–13]. In animals, almost all methylated cytosines occur in the CG context[10]. However, plant cytosines can be methylated in symmetrical CG and CHG (H = A, T, or C) contexts, but at lower levels in the non-symmetrical CHH context[11]. In *Arabidopsis*, DNA methylation in all three contexts are enriched in transposons[14,15]. In short transposable elements (TEs), the 24nt-siRNA targeting DOMAINS REARRANGED METHYLTRANSFERASE 2 (DRM2) maintains CHH methylation, while the CHG methylation and H3K9me2 form a self-reinforcing loop between CHROMOMETHYLASE 3 (CMT3) and KRYPTONITE (KYP)[16–19]. In long TEs, both CMT2 and CMT3 mediate CHG methylation, and CMT2 mediates CHH methylation through binding to H3K9me mark[20]. Strikingly, non-CG methylation almost lost in *drm1, drm2, cmt2, cmt3* (*ddcc*) quadruple mutant, which shows a global increase of RNA-seq reads in heterochromatic regions[20,21], suggesting different non-CG pathways cooperate to silence TEs in the genome.

Within the context of chromatin, there is a complex crosstalk between DNA methylation and histone modifications, especially histone methylation[22]. It was well studied that DNA methylation and histone H3K9 methylation form a self-reinforcing loop to maintain heterochromatic state in *Arabidopsis*[17]. On the other hand, recent work from mammalian system reveals that cytosine methylation impacts binding of transcription factors (TFs), CTCF, and polycomb-like proteins (PCLs), one family of PRC2-associated factors[23], to specific DNA sequence, which may affect transcription states, higher-order chromatin interactions and chromatin states[24,25]. A high-throughput TF-binding site discovery method, namely DNA affinity purification sequencing (DAP-seq), allows to identify the potential genomic-binding sites of several hundreds of TFs[26]. Using this method, they found that 76% of *Arabidopsis* TFs they studied were sensitive to DNA methylation[27]. However, whether and to what extent DNA methylation affects the binding of a transcriptional activating histone-modifying enzyme genome-wide in vivo, especially in plant, is largely unknown.

In this study, we show that non-CG methylation in CTCTGYTY motifs is one way to prevent REF6 targeting. Structural analysis demonstrates that CHG methylation is unfavorable for REF6 binding and attenuates REF6-binding affinity. In vivo chromatin immunoprecipitation (ChIP) coupled with high-throughput bisulfite sequencing (ChIP-BS-seq) result shows that REF6 prefers to bind hypo-methylated DNA and ectopically binds to multiple new targets in *ddcc* quadruple mutant where non-CG methylation is significantly diminished. Our findings not only demonstrate the targeting mechanism of REF6, but also reveal a mechanism for a transcriptional-activating histone-modifying enzyme in avoiding heterochromatic binding through its intrinsic DNA methylation unfavorable DNA-binding activity.

## Results

**REF6 prefers to bind DNA hypo-methylated regions.** Because REF6-bound regions are depleted in heterochromatin regions marked by H3K9me2[7], which is strongly associated with DNA methylation[17–20], and REF6-binding motifs contain non-CG sequence context (CHG and CHH), we hypothesized that DNA methylation affects REF6 binding. To test this hypothesis, we used published whole genome bisulfite sequencing (WGBS) datasets[21] to compare DNA methylation levels in regions containing CTCTGYTY motifs, no matter whether REF6 could bind or not. Although 24,786 CTCTGYTY motif-containing regions not bound by REF6 (REF6−) were evenly distributed throughout the genome, sites of REF6 occupancy (REF6+) were mainly located on chromosome arms (including euchromatin and facultative heterochromatin) and were negatively correlated with hypermethylated regions of constitutive heterochromatin (Fig. 1a and Supplementary Fig. 1a), regardless of CG or non-CG sequence context (Mann–Whitney *U*-test, *P*-value < 2.2e−16) (Fig. 1b, c). In *Arabidopsis*, DNA methylation are highly enriched in ~13.5% of protein-coding genes at CG-context within the coding region, and a depletion of DNA methylation at transcriptional start and termination sites, which are referred to as "gene body methylation" (GbM). Here we found that ~84% of REF6-bound target genes were unmethylated genes (Supplementary Fig. 1b, c) and 8.2% of those were gene-body methylated (Supplementary Fig. 1b). Although the role of mCG in GbM are largely unknown, mCG level is enriched in the transcribed region but depleted in the transcription start site (TSS) and transcription termination site (TTS), which is opposite to REF6-binding pattern (Supplementary Fig. 1d). Furthermore, the cytosines in REF6-bound CTCTGYTY motifs tended to be unmethylated in CHG contexts (Mann–Whitney *U*-test, *P*-value < 2.2e$^{-16}$) (Fig. 1d and Supplementary Fig. 1e). These results show that REF6 prefers CTCTGYTY motifs in which cytosine is unmethylated.

To further confirm that REF6 preferentially bind to unmethylated DNA motifs in vivo, we performed REF6 ChIP-BS-seq[28] in Col, compared with *ddcc*, in which non-CG methylation is completely lost[20,21] (Supplementary Table 1 and Supplementary Fig. 2a). The results showed an anti-correlated profile between REF6-binding signal and DNA methylation level at REF6-binding peaks (Fig. 2a and Supplementary Fig. 2b). DNA methylation level of REF6-binding peaks identified by ChIP-BS-seq in Col are as low as that in *ddcc*, indicating that there is not significant difference between wild-type Col and *ddcc* for differential sensitivity to non-CG DNA methylation (Fig. 2b). Moreover, REF6 bound DNA showed lower methylation level compared to that in WGBS data, indicating that REF6 bound DNA was depleted for DNA methylation while the methylation at REF6-binding sites seen in WGBS data may come from DNA without REF6 binding in some cell types (Fig. 2b). These results give

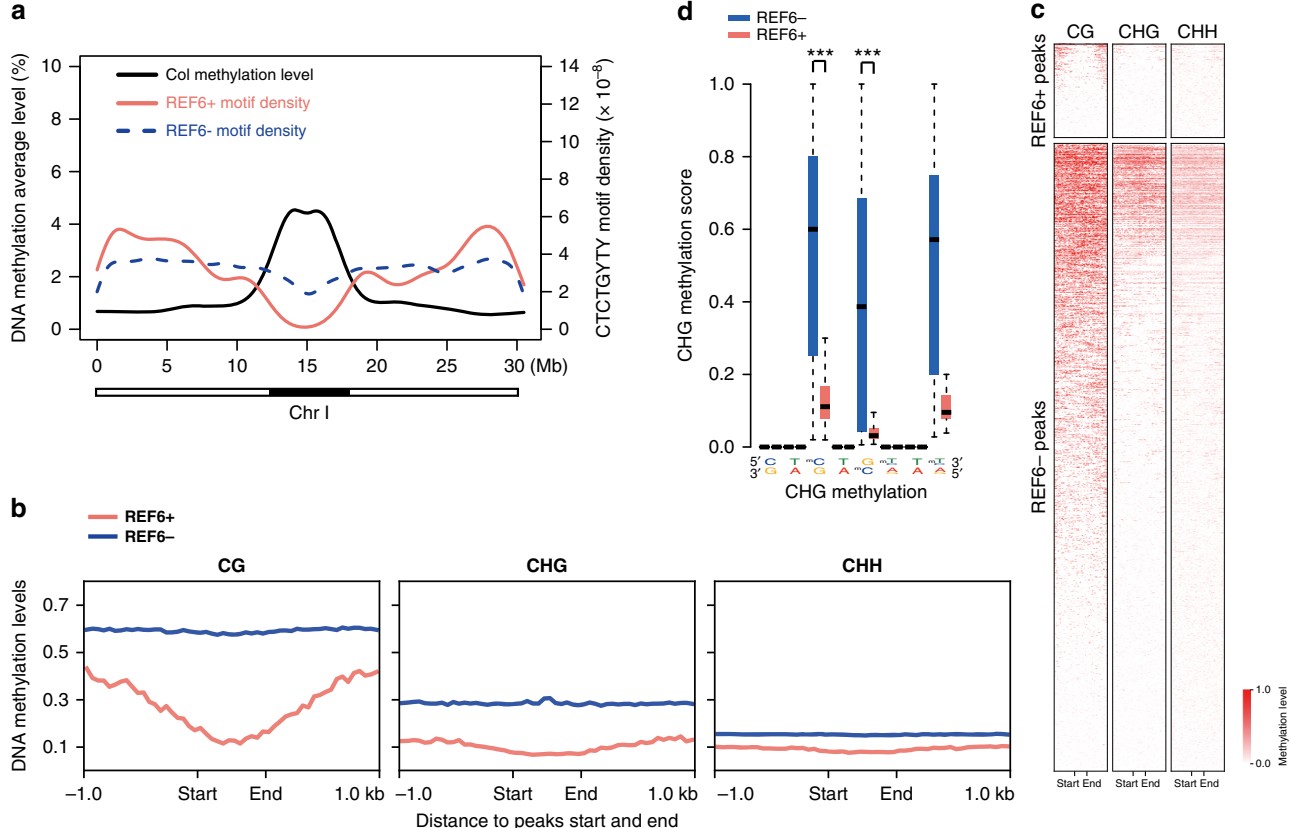

**Fig. 1** REF6 prefers to bind DNA hypo-methylated regions. **a** DNA methylation level, as well as REF6 bound and unbound motif density, on Chromosome I. The black bar below the panel shows the position of heterochromatin. **b** Average distribution of DNA methylation levels (CG, CHG, CHH) over central regions within REF6-bound (REF6+) and REF6-unbound (REF6−) motifs; peak "start" and "end" sites are separated by 600 bp. REF6− regions were constructed as described in "Methods". **c** Heatmap of DNA methylation levels within REF6-bound and REF6-unbound regions. **d** CHG methylation levels at CTCTGYTY motifs in REF6-bound (REF6+) and unbound (REF6−) regions. The box plots display the median (center line) and interquartile range (IQR; from the 25th to 75th percentile), and the whiskers represent the minimum and maximum of DNA methylation score from 0 to 1. Mann–Whitney $U$-test was used to calculate the $P$-value. ***$P < 0.001$

direct evidence supporting REF6 prefer to bind hypomethylated DNA in the *Arabidopsis* genome.

**Cytosine methylation decreases DNA-binding affinity of REF6.** To determine whether and to what extent 5mC repels direct binding of REF6 to CTCTGYTY motifs, we performed electromobility shift assays (EMSA) using 50-bp DNA fragments from the *AT1G02230* and *AT4G11710* genes, both of which contain the CTCTGTTT motif, with or without 5mC. The probes were incubated with recombinant GST-tagged C-terminal REF6 fused to a tandem array of four Cys$_2$-His$_2$ (C2H2)-ZnFs (GST-REF6C, 1239–1360 a.a.). GST-REF6C bound all probes well in the absence of 5mC, as we reported previously[7] (Fig. 3). DNA probes with differential 5mCs on the "top" strand (Fig. 3), including cytosine methylation at position 1 (5mC$_1$, CHH context) and 5mC$_3$ (CHG context), had severely reduced binding affinity, whereas the presence of two 5mCs on the top strand (5mC$_1$+5mC$_3$) and a single 5mC on the bottom strand (5mC$_5$, CHG context) completely abolished the protein–DNA interaction (Fig. 3).

**Crystal structures of REF6 ZnF-clusters and unmethylated DNA.** To determine in greater detail why REF6 binding to methylated cytosine is unfavorable, we solved the crystal structures of REF6 ZnF-clusters bound to double-stranded DNA of *NAC004* containing the CTCTGTTT motif and methylated

DNAs of *NAC004*_5mC$_1$ (ZnF2-4-5mC$_1$) and *NAC004*_5mC$_3$ (ZnF2-4-5mC$_3$) (Fig. 4a and Supplementary Fig. 3). Detailed diffraction statistics were summarized in Supplementary Table 2. ZnF domains adopt the canonical ββα fold, with a small β-sheet packed against a helix in a globular structure, and wrap more than one turn of the DNA double helix, interacting with DNA in the major groove with a classic α-helix (Fig. 4b). Although the overall structures of the REF6 ZnF domains were similar between complexes with methylated or unmethylated DNA oligos, there are minor conformational changes when DNA strands carrying the methyl-group (Fig. 4c, d and Supplementary Fig. 3). In ZnF2-4-5mC$_1$ complex, F1339 contributes some hydrophobic interactions to the binding of the 5mC$_1$, but it could not compensate for the repulsion from the side chain of D1342 due to repellency from the carboxyl group (Fig. 4c and Supplementary Fig. 3a). In ZnF2-4-5mC$_3$ complex, S1312 denotes the repelling force to the methyl-group, decreasing the binding ability to the C3 base (Fig. 4d and Supplementary Fig. 3b). Because REF6-ZnFs have low affinity for 5mC$_5$, we could not obtain a crystal structure of ZnF2-4 with 5mC$_5$ probes. Modeling analysis suggested that the presence of a methyl group at the C5 atom of C$_5$ would sterically obstruct W1311 in the cytosine-specific conformation (Fig. 4e), explaining the diminished binding to the C$_5$-methylated oligo (Fig. 3).

A recent global analysis revealed that cytosine methylation impacts binding of TFs to DNA: hydrophobic interactions promote direct binding, whereas steric hindrance inhibits

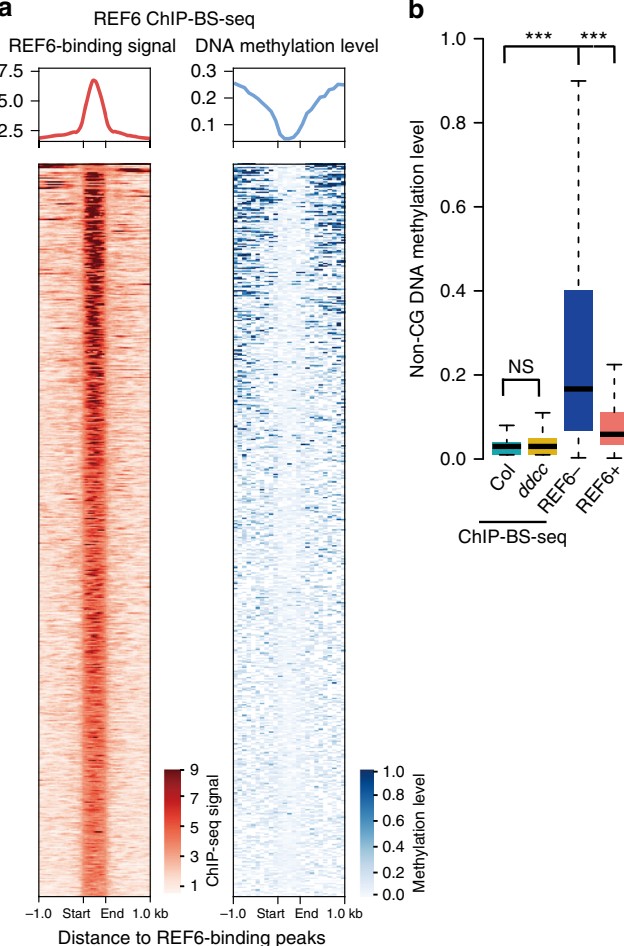

**Fig. 2** REF6 prefer to bind hypomethylated regions in *Arabidopsis* genome. **a** Heat maps of REF6 occupancy and DNA methylation level in 1.0 kb surrounding REF6-binding peaks. **b** Box plots showing average non-CG DNA methylation level of in vivo REF6-binding regions. The DNA methylation levels of REF6-binding regions in Col and *ddcc* are measured by ChIP-BS-seq data, while those in REF6-bound (REF6+) and REF6-unbound (REF6−) regions are measured by whole genome bisulfate sequencing in Col from WGBS data. In vivo REF6-binding regions and REF6+ show significant difference (Mann–Whitney *U*-test, *P* < 2.2e−16) to REF6− motifs. NS, not significant. The median (center line), interquartile range (IQR; from the 25th to 75th percentile) and the ends of whiskers indicating the minimum and maximum of DNA methylation score from 0 to 1, are shown in the boxplots. Mann–Whitney *U*-test was used to calculate the *P* value. ***P* < 0.001

binding[25]. Further analysis by isothermal titration calorimetry (ITC) between REF6-ZnFs and the DNA probes with or without 5mCs revealed that 5mCs lead to great reduction in the binding affinity to the CTCTGYTY motif (Fig. 4f and Supplementary Table 3). In combination with EMSA assay, the biochemical and structural studies demonstrated that DNA methylation at the CTCTGYTY-motif, especially at the CTG core, is sufficient to abolish (or severely attenuate) the affinity of REF6-ZnF for this DNA sequence.

**DNA methylation represses REF6 binding at specific loci.** In contrast to the situation in mammals, non-CG methylation is abundant in heterochromatin regions in *Arabidopsis*. DNA methylation at non-CG context are primarily mediated by DRM1, DRM2, CMT2, and CMT3[10,21,29,30]. Previous work showed that

non-CG methylation in the *Arabidopsis* genome is mostly absent in *drm1 drm2 cmt2 cmt3* (*ddcc*) quadruple mutants[20] (Supplementary Fig. 4). To investigate whether non-CG methylation blocks REF6 targeting in vivo, we profiled the genome-wide localization of REF6 in wild-type Col and *ddcc* mutants by ChIP coupled with high-throughput sequencing (ChIP-seq) (Supplementary Table 1). Two biological replicates of REF6 ChIP-seq with anti-REF6 antibody showed high Pearson correlation coefficient with each other (Supplementary Fig. 5a, b). It revealed that in Col, a total of 2026 of 600 bp REF6-binding peaks covering 1907 genes were bound by REF6, 88% of which have one or more CTCTGYTY motifs; these results were highly correlated (*r* = 0.83) with those REF6-HA ChIP-seq using anti-HA antibody[7] (Supplementary Fig. 5c). Moreover, REF6 target genes were efficiently enriched in Col in comparison with a *ref6* mutant (Supplementary Fig. 5d), indicating that the anti-REF6 antibody worked well.

Within 1220 CTCTGYTY-motifs containing loci of CHG hypomethylated differentially methylated regions (DMR) in euchromatin (from chromosome arm), REF6 exhibited ectopic binding in ~14 loci in *ddcc* mutant with two biological replicates for REF6 ChIP-seq (Fig. 5a, b, Supplementary Fig. 6a, and Supplementary Table 4). Although the number of ectopic-binding peaks of REF6 is relatively low in *ddcc* mutant, the binding affinity to these sites are significantly and reproducibly high (Fig. 5b), indicating these ectopic-binding peaks are bona fide REF6-binding sites in *ddcc* mutant. Loss of non-CG methylation has minor effects on euchromatic structures associated with gene expression in *ddcc* mutant[20]. Here we found that most of these ectopic REF6-binding sites in *ddcc* are located in or neighbored with short TEs in euchromatic regions, and some of these ectopic-binding events are associated with transcriptional activation of TEs or their neighbor protein-coding genes (Supplementary Fig. 6b).

In addition, we validated REF6 binding at these new target sites in Col, *ddcc*, *cmt2*, *cmt3*, and *drm1 drm2* mutants by quantitative PCR (ChIP-qPCR), using an independent batch of samples. Consistent with the ChIP-seq results, REF6 bound to the new target loci in *ddcc* and *cmt3* mutants, but not in *cmt2* or *drm1 drm2* mutants (Fig. 5c and Supplementary Fig. 6c). The ectopic-binding sites were located in restricted regions with low levels of H3K9me2 on chromosome arms (Fig. 5d and Supplementary Fig. 7). Because CMT3 methylates cytosines predominantly in the CHG context, these results indicate that DNA methylation in the CHG context of the CTCTGYTY motif play more important roles in repeling REF6 binding (Fig. 6).

**Discussion**

Taken together, the findings described here demonstrate how DNA cytosine methylation regulates the affinity of a plant H3K27me3 demethylase, REF6, to CTCTGYTY-motif both in vitro and in vivo. It is unclear why REF6 prefers to bind CTCTGYTY-motif in euchromatic regions and depletes from heterochromatic regions[7]. One mechanism, we tested here, is that DNA methylation may directly repress REF6-binding to motifs located in heterochromatic regions and potentially avoids unwanted transcriptional activation in heterochromatin.

Although DNA methylation on CTCTGYTY motif is sufficient to repel REF6-binding in vitro (Figs. 3 and 4), loss of DNA methylation seems necessary but not sufficient for ectopic REF6-binding in vivo. Comparing with DNA-binding proteins in mammalian systems, such as OCT4[25], REF6 only ectopically bind to very limited number of loci even in mutants with dramatic decrease of non-CG DNA methylation. It is speculated that REF6 ectopic-binding sites in *ddcc* mutant result from not only loss of

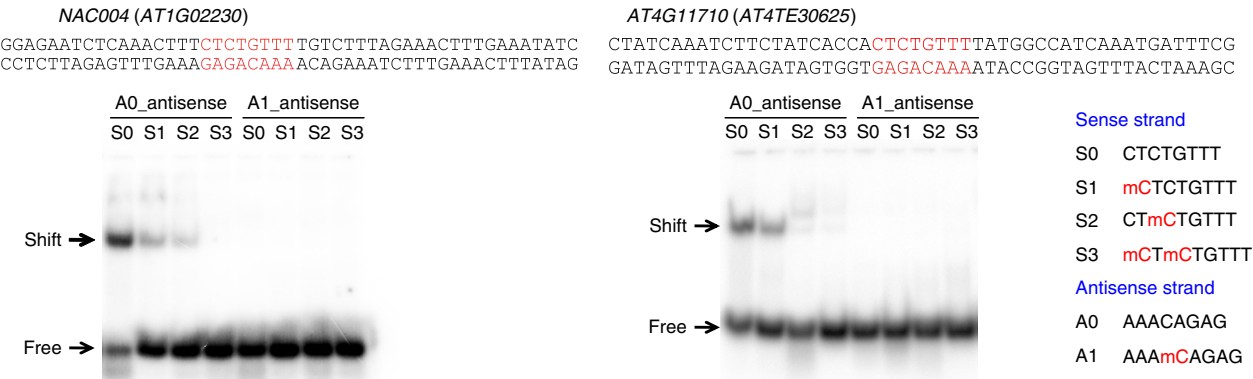

**Fig. 3** Cytosine methylation decreases DNA-binding affinity of REF6-ZnF in vitro. EMSA with *NAC004* (*AT1G02230*) and *AT4G11710* probes. REF6-ZnF specifically bound the unmethylated probes, but had significantly lower (or no) affinity for probe sequences containing one or more methylated cytosines. Source data are provided as a Source Data file

**Fig. 4** Molecular basis of REF6 ZnF-clusters and unmethylated DNA. **a** DNA sequence used for crystal analysis and ITC assay. *NAC004P* is the partial of *NAC004* probe. **b** Crystal structure of REF6 ZnF2-4 in complex with *NAC004* dsDNA. Zinc finger domains are highlighted in blue (ZnF2), lime green (ZnF3), and purple (ZnF4). ZnF2-4 and DNA are shown as cartoon representations. The coding strand of DNA is shown in gray, and the non-coding strand in black. Spheres are Zn atoms. **c** Structural basis of the interaction of mC$_1$ with F1339 and D1342. F1339 engages in hydrophobic interactions with 5mC$_1$ and D1342 forms a weak C–H...O type of hydrogen bond with the 5mC methyl group. **d** Structural basis of interaction of mC$_3$ with S1312 and E1315. S1312 makes a weak C–H...O type of hydrogen bond with the 5mC methyl group and E1315 forms a direct H-bond with mC$_3$. **e** Modeling a methyl group onto unmodified C$_5$ in the non-coding strand reveals potential steric hindrance (indicated by a red star) with W1311. **f** ITC assays showing decreased interaction between REF6-ZnFs and methylated DNA probes. NDB no detectable binding

**Fig. 5** DNA methylation represses REF6 binding at specific loci. **a** REF6 ChIP-seq signal at all regions in wild-type Col and the *ddcc* mutant. The thick red dots represent the region in **b**. **b** Genome-browser view of REF6 binding and DNA methylation in Col and the *ddcc* mutant at the *AT2TE32120* and *AT5TE57090* loci. The *CUC1* locus was used as the control. Gene models from TAIR10 are shown in black at the bottom of the panel. **c** ChIP-qPCR validation of REF6 binding at *AT2TE32120* and *AT5TE57090*, using ChIP samples of another biological replicate, in wild-type Col and the *ddcc*, *cmt2-3*, *cmt3-11*, and *drm1 drm2* mutants. *HB23* and *NC4* were used as positive and negative controls, respectively. ChIP-qPCR was performed in three technical replicates. Error bars indicate mean ± SE from three independent experiments. The individual data points are shown as dots. Source data are provided as a Source Data file. **d** Distribution of REF6 ChIP-seq signal, TEs (red dot), and H3K9me2 density (yellow line) across chromosome 2. Blue lines with arrows indicate ectopic-binding sites in *ddcc* mutants, one of which is shown at the top right corner. Gray shading covering the area of high H3K9me2 density represent the heterochromatin regions

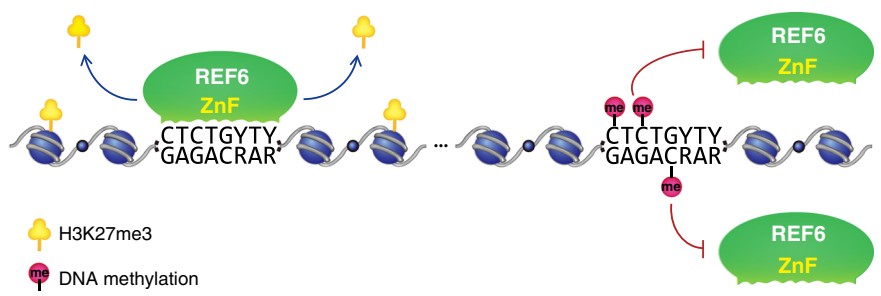

**Fig. 6** Model of how DNA methylation prevents REF6 binding to the CTCTGYTY motif

DNA methylation, but also changes of other chromatin features. Additional factors, such as other epigenetic markers of heterochromatin and higher-order chromatin structure, may prevent REF6 from targeting to heterochromatic regions[7]. It is still unclear what features of specific short TEs in the genome enabling REF6-targeting them in *ddcc* mutant. In future studies, it will be of great interest to further explore how recruitment of a chromatin-modifying enzyme is tightly regulated to achieve the appropriate level of chromatin modification and maintain proper chromatin status at the right place in the genome.

## Methods

**Plant materials**. All mutant lines used in this study were in the Columbia (Col) ecotype background. *ref6-5* (SALK_059549), *cmt2-3* (SALK_012874), *cmt3-11* (SALK_148381), and *drm1-2;drm2-2* (CS16383) mutants were ordered from the *Arabidopsis* Biological Resource Center. The *drm1-2; drm2-2; cmt2-7; cmt3-11* quadruple mutant, described previously[20], was a kind gift from Steve Jacobsen's lab. All *Arabidopsis* materials were grown on half-strength Murashige and Skoog (MS) medium containing 1% sucrose at 22 °C under long-day conditions (LD: 16 h light, 8 h dark), and 10-day-old seedlings were used for all experiments.

**Chromatin immunoprecipitation**. About 3 g of seedlings were collected without crosslinking and stored at −80 °C until use. ChIP was performed as previously described[7] with minor modifications. Briefly, plant tissues were ground to a fine powder in liquid nitrogen and resuspended in 30 ml of ChIP extraction buffer 1 (0.4 M sucrose, 10 mM Tris–HCl, 10 mM MgCl₂, 1 mM dithiothreitol [DTT], 0.1 mM PMSF, protease inhibitor cocktail, pH 8.0). After the powder dissolved, 810 µl of 37% formaldehyde solution was added (final concentration, 1%), and the sample was incubated at 4 °C for 10 min on a rotating mixer to crosslink DNA and protein. The crosslinking reaction was quenched by adding 1.9 ml of 2 M glycine, followed by incubation at 4 °C for 5 min. The nuclear pellet was isolated as described previously[31], and then resuspended with high-salt buffer (20 mM Tris–HCl, 500 mM NaCl, 1 mM EDTA, 1% Triton X-100, 0.1% SDS, pH 8.0) and kept on ice for 30 min before sonication. The samples were sonicated for 12 min (15 s on, 30 s off, for 48 times, high intensity) in a BIORUPTOR (Diagenode UCD-200, Belgium) to yield DNA fragments of 0.2–0.8 kb. Lysates were cleared by centrifugation (16,000×g, 10 min, 4 °C) and diluted with one volume of 20 mM Tris–HCl (pH 8.0) before immunoprecipitation with anti-REF6 antibody (custome mouse monoclonal antibody by Abmart against peptide QEGSDGHEEARDGR). After 5% of the sample was set aside as input, the rest of the supernatant was incubated with antibody-bound Dynabeads Protein G (Life Technologies, 10003D, 30 µl beads bound to 0.4 µg REF6-antibodies according to the user's manual) at 4 °C for 3 h on a rotating mixer. Beads were washed two times for 5 min at 4 °C in low-salt washing buffer (20 mM Tris–HCl, 150 mM NaCl, 1 mM EDTA, 1% Triton X-100, 0.1% SDS, pH 8.0), followed by two washes for 5 min in high-salt buffer, one wash for 5 min in TBST, and one wash in TE (10 mM Tris–HCl, 1 mM EDTA, pH 8.0). DNA elution, reverse-crosslinking, and DNA purification steps were performed as described previously[31].

The ChIP DNA was subjected to qPCR analysis or Illumina sequencing. For ChIP-seq, 1–2 ng DNA was used per sample. Libraries were constructed with the NEXTflex Rapid DNA-seq Prep Kit for Illumina Sequencing (BIOO Scientific, #5144-03). Primers for qPCR are listed in Supplementary Table 5. One intergenic region that is not bound by REF6 was used as a negative control (NC4). Source data are provided as a Source Data file.

**ChIP-bisulfite-Seq**. ChIP with REF6 antibody was performed with Col, *ddcc*, and *ref6-5*, and validation of REF6 binding was confirmed by ChIP-qPCR. Twenty nanograms of ChIP DNA was used for ChIP-BS-seq. Libraries were constructed with NEBNext Ultra II DNA Library Prep Kit for Illumina (NEB, #E7645) together with methylated adaptor (NEB, #E7535S). After adaptor ligation, DNA was treated with bisulfite solution according to EpiTect Fast DNA Bisulfite Kit (QIAGEN, #59824). 0.2 ng λDNA was added to detect bisulfite treatment efficiency[28].

**ChIP-seq analysis**. Paired-end sequencing reads from ChIP-seq were mapped to the *Arabidopsis thaliana* TAIR10 reference genome using Bowtie2[32] (version 2.2.8) with default parameters in local alignment mode. Multiple mapping reads, unmated reads, and mated reads that mapped too far apart (>4×fragment length) for paired-end reads were excluded for downstream analysis. In order to avoid double counting in overlapping regions for paired-end data, the first read mate was used to calculate normalized genome coverage tracks by deepTools2[33] (–normalizeTo1x 1.19e8–extendReads*–binSize 1) after extending ChIP-seq reads to the average estimated fragment length, disregarding the second read mate. The binding intensity of REF6 protein in merged peak regions was estimated by read counts per million mapped reads (RPM) after normalization by library size factor. The density map of reads was then converted to BigWig files and visualized using the integrative genomics viewer[34] (IGV).

ChIP-seq peaks were called using MACS2[35] (v 2.1.1) with the "–gsize 1.19e8–keep-dup 1" option. Peaks were annotated to the gene with the closest TSS using ChIPseeker[36]. To compare different epigenetic states of REF6-binding regions and non-REF6-binding regions that also harbor CTCTGTYTY motifs, we artificially constructed 600-bp regions by sliding a window across the whole genome. The middle point of assumed 600-bp peak was set as the location of the first base of a non-binding CTCTCYTY motif and extended 300 bp in the both 5′ and 3′ directions. Deeptools2[37] was used to create profile plots and heatmaps to display the occupancy signal over sets of genomic regions. MAnorm[38] was used to normalize mapped read counts for regions of interest (merged ChIP-seq-enriched regions) and identify differential regions, using a fold change threshold of 4 and P-value threshold of 0.0001. Scatterplots of two normalized read counts and Pearson correlation coefficients were generated using R. To explore chromatin state surrounding the REF6-binding sites, we defined the heterochromatin regions (Chr1:12,500,000–17,050,000, Chr2:2,300,000–6,300,000, Chr3: 12,800,000–14,800,000, Chr4: 1,620,000–2,280,000; 2,780,000–5,804,000, Chr5: 10,680,000–14,000,000) across chromosomes by considering both H3K9me2 read density[39] and TE size.

**ChIP-bisulfite-Seq and DNA methylation analysis**. ChIP-BS-seq and WGBS reads were both mapped to the TAIR10 genome with BS-seeker2[40] with the setting "-m 2", allowing two mismatches. The methylation level of each base site covering at least four reads were computed as the ratio #C/(#C+#T). Then the enriched regions of ChIP-Bisulfite-Seq were searched using MACS2 with the input of bisulfite reads mapping position information. Similarly, peaks binding signal and methylation level from ChIP-BS-Seq were processed by deepTools2. DMRs were identified as previously described[21] using the R package DMRcaller (http://bioconductor.org/packages/DMRcaller/). The intersection between DMRs and CTCTGTYTY motifs was assumed as the possible differentially binding site in two genotypes. Lists of GbM and unmethylated genes were used as previously reported[41].

**Electrophoretic mobility shift assay (EMSA)**. The REF6C fragment (encoding amino acids 1239–1360 including the stop codon) was cloned into pGEX-6p-1 (GE Healthcare), expressed in *E. coli* (BL21 codon plus, Stratagene), and purified using Glutathione Sepharose 4B beads (GE Healthcare) as described previously[7]. EMSA was performed as described with minor modifications[7]. Complementary oligonucleotides with or without m5C modifications were annealed and 5′-end labeled with α-³²P-dATP using T4-PNK (NEB, M0201). About 100 ng of GST-REF6C protein and 3 nM ³²P-labeled probes were incubated in 10 µl reaction mixture (containing 25 mM Tris–HCl, 100 mM NaCl, 2.5 mM MgCl₂, 0.1% CA-630, 10% glycerol, 1 µM ZnSO₄, and 1 mM DTT, pH 8.0) for 1 h on ice, and then separated in 6% native polyacrylamide gel in 0.5X TBE buffer (40 mM Tris–HCl, 45 mM boric acid, 1 mM EDTA, pH 8.3) at 80 V for about 80 min (room temperature). Source data are provided as a Source Data file.

**Recombinant protein expression and purification**. cDNA fragments encoding the ZnF2-4 domain (residues 1260–1360) and ZnF1-4 domain (residues 1239–1360) of *A. thaliana* REF6 protein were PCR-amplified and cloned into pGEX-6P-1 (GE Healthcare). Plasmids were transformed into *E. coli* BL21 (DE3) (Stratagene). Bacterial cells were cultured at 37 °C in LB medium; ZnCl₂ was added to a final concentration of 150 µM. Expression of recombinant proteins was induced by addition of 0.2 mM isopropyl β-D-thiogalactopyranoside and incubation at 18 °C overnight. Bacteria were harvested and lysed with a high-pressure cell cracker in lysis buffer containing 20 mM Tris–HCl, pH 7.5, 250 mM NaCl. After centrifugation, the cleared extract was incubated with glutathione sepharose 4 fast flow beads (GE Healthcare). GST fusion proteins were eluted with 20 mM Tris–HCl (pH 7.5), 250 mM NaCl, and 10 mM glutathione, and then loaded onto a HiTrap-SP column (GE Healthcare). The GST tag bound to the SP column was removed using PreScission protease (purified in-house). Protein was further purified on HiTrap SP columns and a HiLoad Superdex 75 16/60 column (GE Healthcare), and concentrated to 26 mg/ml in 20 mM Tris–HCl (pH 7.5), 300 mM NaCl.

**Isothermal titration calorimetry**. ITC measurements for DNA binding were performed at 25 °C on an iTC200 calorimeter (Microcal). REF6 ZnF1-4 (residues 1239–1360, without the GST tag) and DNA substrates were dialyzed in 20 mM Tris–HCl (pH 7.5), 150 mM NaCl, 20 µM ZnCl₂. The DNA concentration in the cell was 16 µM, and the protein concentration in the injection syringe was 200 µM. Data were analyzed using the MicroCal ORIGIN software with a single-site-binding model.

**Crystallography**. Before crystallization, purified proteins (10 mg/ml) were incubated with annealed oligonucleotides at a molar ratio of 1:1.2 for 0.5 h at room temperature. Crystals were obtained by the hanging-drop method. Crystals of ZnF2-4-*NAC004* were grown in 12% PEG 3350, 0.2 M NH₄F. Crystals of ZnF2-4-*NAC004*_5mC₁ were grown in 12% PEG 3350, 0.15 M malic acid. Crystals of ZnF2-4-*NAC004*_5mC₃ were grown in 16% PEG 3350, 0.03 M citrate acid, 0.07 M bis–tris propane (pH 7.6). All crystals grew within 1 day at 18 °C.

Crystals were flash-frozen by plunging into liquid nitrogen. X-ray diffraction data were collected at beamline BL-19U1 of the Shanghai Synchrotron Radiation Facility (SSRF). HKL3000 was used for diffraction data processing[42]. The structure of ZnF2-4 was solved by molecular replacement with the coordinates of 4R2S as the model, and the other crystal structures were solved by molecular replacement with the coordinates of ZnF2-4, using the PHENIX[43] and Phaser programs[44]. All structural models were refined with REFMAC5 in the CCP4 package[45,46] and adjusted using Coot[47]. Model representations were all finished using PyMOL (DeLano Scientific LLC).

**Reporting summary**. Further information on research design is available in the Nature Research Reporting Summary linked to this article.

## Data availability

X-ray structures (coordinates and structure factor files) of REF6 ZnFs with bound DNA have been submitted to PDB under accession numbers 6JNL, 6JNN, 6JNM represents for ZnF2-4-NAC004, ZnF2-4-NAC004_5mC₁, ZnF2-4-NAC00_5mC₃, respectively. ChIP-seq data sets generated in this study have been deposited in the Gene Expression Omnibus (GEO) under accession GSE111830. The source data underlying Fig. 3, Fig. 5c, Supplementary Fig. 5d, and Supplementary Fig. 6c are provided as a Source Data file.

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

## Acknowledgements

We thank Steve Jacobson from UCLA for providing the *ddcc* mutants. We thank the *Arabidopsis* Biological Resource Center for providing T-DNA insertion lines. This work was supported by the National Natural Science Foundation of China (grants 31788103 to X.C., 31230041 to J.M., and 31770323 to X.D.); the Chinese Academy of Sciences (Strategic Priority Research Program XDB27030201 and QYZDY-SSW-SMC022 to X. C.); the National Key Research and Development Program of China (2018YFA0107001 to F.L.); the Youth Innovation Promotion Association of CAS (2018131 to X.D.), and the State Key Laboratory of Plant Genomics.

## Author contributions

Q.Q. and X.C. conceived and designed the study. Q.Q., X.D., K.H., B.W., and Q.Y. performed most of the experiments. High-throughput sequencing data were analyzed by H.M., K.H., and J.Z. performed the ChIP-BS-seq experiments. Q.Q., H.M., X.D., K.H., B. W., Q.Y., F.L., J.M., and X.C. interpreted the data. Q.Q., X.D., J.M., and X.C. wrote the paper.

## Additional information

**Competing interests:** The authors declare no competing interests.

