## [Peer Review File · Nature Communications]

Reviewers' comments:

Reviewer #1 (Remarks to the Author):

This paper describes how methylcytosines can repel binding of the histone demethylase REF6. The evidence to support these conclusions is from 1) the depletion of REF6 binding to methylated motifs from genome-wide ChIP data, 2) EMSA on two target loci, 3) binding constants from ITC and 4) ectopic binding in a mutant that has no nonCG methylation.

A recent publication by the Ecker Lab showed that ~100 DNA binding proteins in Arabidopsis are sensitive to DNA methylation using the newly developed DAP and ampDAP-seq approaches (O'Malley et al, Cell, 2017). Therefore, the results presented in this study represent an incremental advance given data are only presented for a single DNA binding protein, REF6.

The data in this study suggests that although DNA methylation is associated with differential binding it is clearly not deterministic. In the REF6 binding data there are ~15% of binding events that are methylated (line 108). This shows that REF6 can bind methylated DNA.

One key experiment missing would be to perform ChIP-BS-seq using REF6 in Col vs ddcc. How many methylated reads are identified in the Col-0 data. This would be a direct in vivo measurement of differential sensitivity to DNA methylation.

I credit the authors for performing the key experiment, which is to perform ChIP-seq of REF6 in the ddcc mutant. Unfortunately, the results do not support that DNA methylation is the major deterministic reason for REF6 binding as 1198/1220 or 98.2% of peaks did not exhibit ectopic binding whereas only 1.8% did. As shown in Figure 4a, the vast majority of ectopic binding events are the weakest binding events given their position in the plot (bottom left in upper quadrant). This further weakens the importance of the methylated sites being important to controlling REF6 binding activity. Why is the number of sites only reported for one arm of one chromosome? They should be reported for all possible sites.

Furthermore, the rare ectopically bound sites in the ddcc mutant could be due to indirect effects. For example, new regions of open chromatin could make it easier for REF6 to bind DNA or differential chromatin states could also facilitate REF6 binding. Therefore, the ddcc data in this case doesn't provide enough ectopic events to convince the reader that DNA methylation is the causal region for differential binding.

Overall, if there are only few ectopically bound REF6 sites in the ddcc mutant, which loses all nonCG methylation, then what is the importance of this mechanism given that it is already proven that some DNA binding proteins are sensitive to DNA methylation levels. In conclusion, it is of my opinion that the results are an incremental advance that will be of interest to a highly specialized audience.

As an aside, there are also thousands of K27me3 regions in the genome that are differential K27 methylated in different cell types and due to different environments. If REF6 binds the same motif why are all regions not demethylated at the same time since they all possess the same motif? It likely doesn't happen because the REF6 motifs are not the only factor that determines demethylation. There must be some other sequence specific factors that mediate REF6's recruitment to target genes.

Reviewer #2 (Remarks to the Author):

The manuscript by Qiu et al. investigated the potential role of DNA methylation in preventing heterochromatin binding of REF6, a JmjC-domain-containing H3K27me3 histone demethylase. This

is a further extension of previous work from the authors' lab demonstrating that REF6 acts as an H3K27me3 histone demethylase (Lu et al., *Nature genetics*, 2011) and recognizes the CTCTGYTY DNA sequences specifically in the euchromatic regions, but not in the heterochromatin (Cui et al., *Nature genetics*, 2016). In this study, the authors showed that REF preferred to bind unmethylated DNAs both in vitro and in vivo and concluded that DNA methylation (particularly CHG methylation) repels REF6 targeting. The data presented are interesting and will provide a potentially mechanistic insight into the interplay between DNA and histone methylation in plants. The topic is important and should have a broad appeal to the readers of *Nature Communications*. The following are specific comments to further improve this manuscript.

My major concern is the lack of sufficient in vivo data supporting the key conclusion that DNA methylation repels REF6 chromatin targeting. While the in vitro EMSA binding and crystal structural analyses are solid, the supporting in vivo data is weak. The authors performed a correlative data analysis and found that REF6 binding sites are negatively correlated with DNA methylation particularly in the heterochromatin (Figure 1). This is not surprising given the euchromatic enrichment nature of REF6. It is important to compare the CTCTGYTY motifs only in the euchromatin with and without REF6 binding. The authors should also provide more details regarding the data analysis. For example, within the 3464 REF6 binding regions, how many of them contain CTCTGYTY motifs? What about the number of CTCTGYTY motifs without REF6 binding? Also, it is beneficial to the general readers to provide more explanations on the purposes and results of the genomic analysis.

Along the same line, my another concern is the biological significance of the anti-correlation between heterochromatic DNA methylation and REF6 binding. Since REF6 is a H3K27me3 demethylase and H3K27me3 is mainly in the body of euchromatic genes. It doesn't make too much sense for me that the authors draw strong conclusion based on the correlation from the DNA methylation in heterochromatin, unless the authors can demonstrate a distinct role of REF6 in heterochromatin independent of its H3K27me3 demethylase activity. Thus, it is critical to only focus the analysis on the biological relevant regions. The authors had previously generated H3K27me3 ChIP-seq in ref6 mutants and generated a metaplot in Fig S1d. It is unclear what the purpose and how this metaplot is used in supporting their conclusion. It is important to correlate the hyper H3K27me3 regions in ref6 with the CTCTGYTY motifs for further analysis.

Investigating the REF6 genome-wide occupancy in ddcc mutants is very nice (figure 4). The authors noted ectopic binding of REF6 to short TEs. But, why the number (22 out of 1220) is so low? Are 1220 the total REF6 ectopic binding peaks? Are they mostly in the heterochromatin? Of the 98% (~1200) REF6 ectopic binding peaks that don't overlap with TEs, what are they and what about their DNA methylation levels?

I noticed that the REF6 ChIP-seq signals appear to be general higher in ddcc compared to Col (e.g. CUC1 control loci in Fig. 4b). The details regarding the REF6 ChIP-seq reads and replicates are missing. It is unclear whether the REF6 ChIP-seq has been biologically repeated, which is important as a certain variation in peak identification is expected between replicates. It is more critical to perform extensive normalization analysis for ChIP-seq data to normalize the overall signal to noise ratio between ddcc and Col samples. Although not required for this manuscript, this reviewer highly recommended normalizing across samples using spiked-in methods.

As stated above that REF6 is a H3K27me3 demethylase and H3K27me3 is mainly in the gene body regions, mostly with CG methylation. In Fig.1b, the CG DNA methylation level had a most obvious change among the three DNA methylation contexts. Thus, it makes more sense to test the role of CG methylation in REF6 binding (e.g REF6 ChIP-seq in met1 or ddm1 mutants). The authors may also consider (not required) perform REF6 ChIP-seq in imb1 or ros1dme1dme2 triple mutants (hypermethylation mutants) to provide more evidence.

The manuscript is well written in general, but may benefit to have more details in the results as

well as the method sections and more discussions. It is also important to include a table describing the genomic sequencing data.

L156, change "significant" to "great" as no significant test has been done in the ITC experiments.

Reviewer #3 (Remarks to the Author):

In the current submission "DNA methylation repels REF6 targeting in Arabidopsis" Qiu et al. demonstrate that the Zinc Finger domain of this H3K27 demethylase is sensitive to DNA methylation both in vitro and in vivo and reveal the structural basis for this sensitivity. Furthermore, they demonstrate that by altering the DNA methylation landscape in the ddcc mutant background, REF6 localizes to new target sites. While there is a small, but growing number of TFs shown to be sensitive to DNA methylation based on in vitro analyses, there remains little evidence regarding the effects of such sensitivities in vivo. The data by Qiu et al represent an excellent in vivo example in which DNA methylation affects chromatin association and provide what may be the first example wherein a methylation sensitive DNA binding module connects two epigenetics modifications (DNA methylation and H3K27 methylation). The findings of this manuscript are well supported by the data and I only have minor comments (see below).

minor comments:

line 63, the sentence about the role of DNA methylation in gene silencing is redundant with the second sentence of the same paragraph and could be removed.

As part of the last paragraph of the introduction the authors should also mention the recent work from the Ecker lab in looking at the methylation sensitivity of many TFs via DAPseq assays.

The phrase "intrinsic DNA methylation unfavorable DNA binding activity" is quite awkward. Perhaps "methylation sensitive DNA binding activity" would be better?

The authors use two different nomenclatures to refer to methylated cytosines (mC and 5mC). As these are the same thing, the authors should stick to a single naming system to avoid unnecessary confusion.

In Figure 3e, I believe that the C5 and G5 labels are switched.

line 164-65 the names of DRM2 and CMT3 were already defined in the introduction and thus do not need to be redefined here.

Information regarding the locations of the REF6 peaks from the ChIP experiments (wt, ddcc, etc) should be provided as well as general information about the ChIP libraries (coverage, mapping etc).

If available additional information regarding the effects of the new REF6 targets sites in the ddcc background, like H3K27me levels or gene expression changes, would make an excellent addition to this manuscript.

In figure 4a, the legend refers to "the thick red dot" but there are many red dots.

In figure 4d, the legend refers to chromosome 1 as "chromosome I".

Reviewer #4 (Remarks to the Author):

In this paper, Qiu et al report an "antagonistic mechanism" between REF6 and DNA methylation.

This interesting finding is supported by structural and seq-based genomic profiling studies. Overall, this story is conceptually important and the regulatory mechanism is interesting. I would like to recommend publication of this manuscript if the following concerns are properly addressed.

Major point:

1) According to ITC assays (Figure 3f), the DNA methylation only results in about twofold reduction of the binding affinity. REF6 binds to methylated DNA strands at $K_D=77-173$ nM, which is still a very strong binding event. In this case, I do not hold the view that "5mCs lead to significant reduction in the binding affinity to the CTCTGYTY motif" in vitro. According to the EMSA data (Figure 2), it is hypermethylation of cytosine but not single site 5mC that functions to repel REF6. Therefore, the authors should carefully revisit their conclusion and squarely conclude their observations. An ITC titration using hypermethylated DNA substrate should be performed. If proven true, it's better to change the current title to "DNA hypermethylation repels REF6 targeting in Arabidopsis". Similarly, the hypermethylation state of the "CTCTGYTY" motif should be examined and confirmed in vivo. This can be explored by single-base-resolution sequencing analysis of the genome DNA.

2) Additional perturbation studies should be performed. For example, based on the structural analysis, the authors should be able to design REF6 mutants that can tolerate 5mC. It would be nice to investigate the functional impact if the mutant REF6 is introduced in plant. Such an effort will significantly improve the quality of the story.

Minor points:

1) A complete ITC fitting parameters should be provided.

2) Figure 3e, C5 and G5' are mislabeled. In addition, a role of W1311 should be confirmed by mutagenesis studies.

3) The proposed mechanistic module in Figure 5 is not supported by the present data. In fact, the binding affinity of REF6 to methylated CTCTGYTY motif is not weak, and even corresponding complex crystal structures have even been determined. The red blocking arrow does not properly reflect the experimental observations.

4) In supplementary Table 1, the unit cell angles should be fixed to integral numbers, e.g. (90, 90, 90). These values are not measured ones for the current space groups. Also please double check the space group of ZnF2-4-NAC004-5mC1. The unit cell angles of (90, 90, 120) is not consistent with the P4(3) space group.

Reviewers' comments:

Reviewer #1 (Remarks to the Author):

This paper describes how methylcytosines can repel binding of the histone demethylase REF6. The evidence to support these conclusions is from 1) the depletion of REF6 binding to methylated motifs from genome-wide ChIP data, 2) EMSA on two target loci, 3) binding constants from ITC and 4) ectopic binding in a mutant that has no nonCG methylation.

A recent publication by the Ecker Lab showed that ~100 DNA binding proteins in Arabidopsis are sensitive to DNA methylation using the newly developed DAP and ampDAP-seq approaches (O'Malley et al, Cell, 2017). Therefore, the results presented in this study represent an incremental advance given data are only presented for a single DNA binding protein, REF6.

Response: We thank the reviewer for pointing out this issue. The DAP-seq is a great method to identify potential TF-binding sites. Nevertheless, DAP-seq is an *in vitro* method using recombinant TFs to pull-down purified DNA, which doesn't reflect the binding affinity *in vivo*. Comparing the DAP-seq and ChIP-seq binding sites of the same TFs, we found the DAP-seq tend to get much more binding peaks than ChIP-seq (data not shown). Therefore, we believe that the DAP-seq data cannot fully reflect TFs-binding features *in vivo*. We use both *in vivo* and *in vitro* methods to demonstrate how DNA methylation affects REF6 recruitment, which is of good interest in understanding the interplay between DNA methylation and H3K27me3 beyond TF binding specificity along.

The data in this study suggests that although DNA methylation is associated with differential binding it is clearly not deterministic. In the REF6 binding data there

are ~15% of binding events that are methylated (line 108). This shows that REF6 can bind methylated DNA.

Response: We thank the reviewer to point this out. In *Arabidopsis*, DNA methylation are highly-enriched in ~13.5% of protein-coding genes at CG-context within the coding region and a depletion of DNA methylation at transcriptional start and termination sites, which are referred to as “gene body methylation” (GbM) (Bewick et al., *PNAS*, 2016; Zhang et al., *Cell*, 2006). Here we found that ~84% of REF6-bound genes are unmethylated genes and 8.2% of those are gene-body methylated. Although the role of mCG in GbM are largely unknown, mCG level is enriched in the transcribed region but depleted in the TSS and TTS, which is opposite to REF6-binding pattern (Supplementary Fig. 1d). Therefore, it is not a conflict result that REF6 could bind to the gene body methylated genes. The remaining 7.8% of REF6 bound regions are annotated as non-coding regions with low methylation level as shown in Fig. 1c.

Supplementary Figure 1. (d) Average profiles of REF6 ChIP-seq and DNA bisulfite sequencing signal at gene-body methylated (GbM) genes around the transcription start site (TSS) and transcription termination site (TTS).

One key experiment missing would be to perform ChIP-BS-seq using REF6 in Col vs *ddcc*. How many methylated reads are identified in the Col-0 data. This would be a direct in vivo measurement of differential sensitivity to DNA methylation.

Response: This is a great suggestion to improve our manuscript. To further

confirm that REF6 preferentially bind to unmethylated DNA motifs *in vivo*, we performed REF6 ChIP-bisulfite-sequencing (ChIP-BS-seq) (Statham, *et al. Genome Research. 2012*) in Col, compared with *ddcc*, in which non-CG methylation is completely lost (Supplementary Table 1 and Supplementary Fig. 2a). The results showed an anti-correlated profile between REF6 binding signal and DNA methylation level at REF6 binding peaks (Fig. 2a and Supplementary Fig. 2). DNA methylation level of REF6 binding peaks identified by ChIP-BS-seq in Col are as low as that in *ddcc*, indicating that there is not significant difference between wild-type Col and *ddcc* for differential sensitivity to non-CG DNA methylation (Fig. 2b). Moreover, REF6 bound DNA showed lower methylation level compared to that in WGBS data, indicating that REF6 bound DNA was depleted for DNA methylation while the methylation at REF6 binding sites seen in WGBS data may come from DNA without REF6 binding in some cell types (Fig. 2b). These results give direct evidence supporting that REF6 prefers to bind hypomethylated DNA in the *Arabidopsis* genome.

Figure 2. ChIP-BS-seq shows REF6 prefers to bind hypomethylated regions in *Arabidopsis*

genome.

(a) Heat maps of REF6 occupancy and DNA methylation level in 1.0 Kb surrounding REF6 binding peaks.

(b) Box plots showing average non-CG DNA methylation level of *in vivo* REF6 binding regions. The DNA methylation levels of REF6 binding regions in Col and *ddcc* were measured by ChIP-BS-seq data, while those in REF6-bound (REF6+) and REF6-unbound (REF6-) regions were measured by whole genome bisulfate sequencing in Col from public data. *In vivo* REF6 binding regions and REF6+ show significant difference (Mann-Whitney U test, $P < 2.2e-16$) to REF- motifs. NS, not significant.

Supplementary Figure 2. (b) Average profiles and heatmaps of CG, CHG and CHH methylation level measured by ChIP-BS-Seq in 1 Kb surrounding REF6 binding peaks in Col.

I credit the authors for performing the key experiment, which is to perform ChIP-seq of REF6 in the *ddcc* mutant. Unfortunately, the results do not support that DNA methylation is the major deterministic reason for REF6 binding as 1198/1220 or 98.2% of peaks did not exhibit ectopic binding whereas only 1.8% did. As shown in Figure 4a, the vast majority of ectopic binding events are the weakest binding events given their position in the plot (bottom left in upper quadrant). This further weakens the importance of the methylated sites being important to controlling REF6 binding activity.

Response: We thank the reviewer to be cautious about this point. To prove the reproducibility of identified ectopic binding sites, we performed another biological replicate of ChIP-seq experiments, and checked the binding intensity by RPM (reads per million) value in overlapping regions between two replicates after normalization by library size. Although the number of ectopic binding peaks of REF6 is relatively low in the *ddcc* mutant, the binding affinity to these sites are significantly and reproducibly high in *ddcc* mutant (Fig. 5b), indicating that these ectopic binding peaks are *bona fide* REF6 binding sites in *ddcc* mutant. Our interpretation is that REF6 targeting mechanism must be fine-tuned, and DNA methylation is not the only factor affecting REF6 binding. Additional factors, such as other epigenetic modifications, histone variants, nucleosome positioning, higher-order chromatin structure, or cooperation with other transcription factors, may also participate in REF6 targeting to specific regions.

Figure 5. (b) Genome-browser view of REF6 binding and DNA methylation in *Col* and the *ddcc* at the *AT2TE32120* and *AT5TE57090* loci. The *CUC1* locus was used as the control.

Why is the number of sites only reported for one arm of one chromosome? They should be reported for all possible sites.

Response: We apologize for not describing clearly enough in the text. We showed the information of chromosome 1 in Fig. 5d, and the rest of chr.2-5 were included in previous version Supplementary Fig. 6 (current Supplementary Fig. 7).

Supplementary Figure 7. REF6-binding sites in Col and the *ddcc* mutant across Arabidopsis chromosomes.

(a–d) Distribution of REF6 ChIP-seq signal, TEs (red dot) by size (kb), and H3K9me2 density (yellow line) across chromosomes 1, 3, 4 and 5. Blue lines with arrows indicate ectopic binding sites in the *ddcc* mutant, one of which is shown as an expanded view in the top right corner of each panel. Gray shading covering the area of high H3K9me2 density represent the heterochromatin regions.

Furthermore, the rare ectopically bound sites in the *ddcc* mutant could be due to indirect effects. For example, new regions of open chromatin could make it easier for REF6 to bind DNA or differential chromatin states could also facilitate REF6 binding. Therefore, the *ddcc* data in this case doesn't provide enough ectopic events to convince the reader that DNA methylation is the causal region for differential binding.

Response: We thank the reviewer to point this out. We previously showed that REF6 binding affinity was affected by chromatin states, and open chromatin with exposed DNA sequence tended to promote REF6 binding. We believe that other epigenetic modifications may also participate in REF6 targeting to specific regions which causes the relatively low number of

ectopic binding in *ddcc* mutant. However, we do believe that the *ddcc* data demonstrated that DNA methylation is the cause of differential REF6 binding at some of the regions in the genome. As this can be recaptured in the *in vitro* EMSA result showing that REF6-ZnF can directly bind to *AT4G11710* in the absence of DNA methylation, but cannot bind to methylated *AT4G11710* (Fig. 3). Taking together these *in vitro* and *in vivo* data, we believe that DNA methylation is the cause for differential binding of REF6 at some of the loci, although the number is small.

Figure 3. Cytosine methylation in CTCTGYTY motifs decreases DNA-binding affinity of REF6-ZnF *in vitro*.

EMSA with *AT1G02230* and *AT4G11710* probes. REF6-ZnF specifically bound the unmethylated probes, but had significantly lower (or no) affinity for probes containing one or more methylated cytosines.

Overall, if there are only few ectopically bound REF6 sites in the *ddcc* mutant, which loses all nonCG methylation, then what is the importance of this mechanism given that it is already proven that some DNA binding proteins are sensitive to DNA methylation levels. In conclusion, it is of my opinion that the results are an incremental advance that will be of interest to a highly specialized audience.

Response: We thank the reviewer to point this out. REF6 is the major player in removing Polycomb repressive marks, H3K27me3. Here, we provide evidence and mechanism that REF6 can only function outside heterochromatin due to DNA methylation, revealing a clear interplay

between two major epigenetic marks which we believe will be of general interest in the field of epigenetics. *Arabidopsis* has relatively simple heterochromatin and small amount of DNA methylation which might be the reason why the number is relatively small. However, we believe the mechanism is conserved in other species with more complex heterochromatin.

As an aside, there are also thousands of K27me3 regions in the genome that are differentially K27 methylated in different cell types and due to different environments. If REF6 binds the same motif why are all regions not demethylated at the same time since they all possess the same motif? It likely doesn't happen because the REF6 motifs are not the only factor that determines demethylation. There must be some other sequence specific factors that mediate REF6's recruitment to target genes.

Response: We thank the reviewer to point this out. REF6 is not the only H3K27me3 demethylase in *Arabidopsis*. In addition, H3K27me3 is determined by the equilibration of methyltransferases and demethylases. Therefore, REF6 is not responsible for all the H3K27me3 demethylation events during development. In our opinion, REF6 is mainly important for maintaining low H3K27me3 level of specific loci across many different cell types rather than dynamic removal of H3K27me3 marker during development.

It's possible that some other sequence specific factors affect REF6-binding affinity to specific genes (Yan, *et al. Nature Plants*, 2018). However, our previous work shows that most of REF6 binding sites contain at least one CTCTGYTY-motif and the binding affinity correlated with the number of CTCTGYTY-motif in peak regions (Cui *et al, Nature Genetics*, 2016). We believe CTCTGYTY-motif is the major factor in REF6-targeting, although

there are many other mechanisms fine-tuning the binding of REF6, including DNA methylation as described in this manuscript.

Reviewer #2 (Remarks to the Author):

The manuscript by Qiu et al. investigated the potential role of DNA methylation in preventing heterochromatin binding of REF6, a JmjC-domain-containing H3K27me3 histone demethylase. This is a further extension of previous work from the authors' lab demonstrating that REF6 acts as an H3K27me3 histone demethylase (Lu et al., Nature genetics, 2011) and recognizes the CTCTGYTY DNA sequences specifically in the euchromatic regions, but not in the heterochromatin (Cui et al., Nature genetics, 2016). In this study, the authors showed that REF preferred to bind unmethylated DNAs both in vitro and in vivo and concluded that DNA methylation (particularly CHG methylation) repels REF6 targeting. The data presented are interesting and will provide a potentially mechanistic insight into the interplay between DNA and histone methylation in plants. The topic is important and should have a broad appeal to the readers of Nature Communications. The following are specific comments to further improve this manuscript.

My major concern is the lack of sufficient in vivo data supporting the key conclusion that DNA methylation repels REF6 chromatin targeting. While the in vitro EMSA binding and crystal structural analyses are solid, the supporting in vivo data is weak. The authors performed a correlative data analysis and found that REF6 binding sites are negatively correlated with DNA methylation particularly in the heterochromatin (Figure 1). This is not surprising given the euchromatic enrichment nature of REF6. It is important to compare the CTCTGYTY motifs only in the euchromatin with and without REF6 binding.

Response: We thank the reviewer to point this out. The scientific question of this paper is why REF6 prefers to bind euchromatic regions, even there are plenty of

CTCTGYTY motifs in heterochromatic regions. One mechanism, which we tested in this paper, is DNA methylation may directly repress REF6-binding to CTCTGYTY motifs located in heterochromatic regions. As we shown here, REF6 shows ~14 ectopic binding sites at several short-TEs, rather than binds to the heterochromatic regions in the *ddcc* mutant, in which DNA methylation dramatic decreases in most of the CTCTGYTY motifs (non-CG context).

In euchromatin, we found CTCTGYTY motif of REF6 binding regions are more enriched in chromosome arm, while CTCTGYTY motif without REF6 binding are uniformly distributed. In consistent with previous work, we conclude that motif density is a key factor for the recruitment of REF6 in euchromatin. In addition to motif density, we infer other factors, such as nucleosome positioning, may also have effect on REF6 binding which is out of the scope of this manuscript.

We thank the reviewer in pointing out that the *in vivo* evidence is weak in the previous version. Therefore, to further confirm that REF6 preferentially bind to unmethylated DNA motifs *in vivo*, we performed REF6 ChIP-bisulfite-sequencing (ChIP-BS-seq) (Statham, *et al. Genome Research.* 2012) in Col, compared with *ddcc*, in which non-CG methylation is completely lost (Supplementary Table 1 and Supplementary Fig. 2a). The results showed an anti-correlated profile between REF6 binding signal and DNA methylation level at REF6 binding peaks (Fig. 2a and Supplementary Fig. 2). DNA methylation level of REF6 binding peaks identified by ChIP-BS-seq in Col are as low as that in *ddcc*, indicating that there is not significant difference between wild-type Col and *ddcc* for differential sensitivity to non-CG DNA methylation (Fig. 2b). Moreover, REF6 bound DNA showed lower methylation level compared to that in WGBS data, indicating that REF6 bound DNA was depleted for DNA methylation while the methylation at REF6 binding sites seen in WGBS data may come from DNA without REF6 binding in some cell types (Fig. 2b). These

results give direct evidence supporting that REF6 prefers to bind hypomethylated DNA in the *Arabidopsis* genome.

Figure 2. ChIP-BS-seq shows REF6 prefers to bind hypomethylated regions in *Arabidopsis* genome.

(a) Heat maps of REF6 occupancy and DNA methylation level in 1.0 Kb surrounding REF6 binding peaks.

(b) Box plots showing average non-CG DNA methylation level of *in vivo* REF6 binding regions. The DNA methylation levels of REF6 binding regions in Col and *ddcc* were measured by ChIP-BS-seq data, while those in REF6-bound (REF6+) and REF6-unbound (REF6-) regions were measured by whole genome bisulfate sequencing in Col from public data. *In vivo* REF6 binding regions and REF6+ show significant difference (Mann-Whitney U test, $P < 2.2e-16$) to REF- motifs. NS, not significant.

The authors should also provide more details regarding the data analysis. For example, within the 3464 REF6 binding regions, how many of them contain CTCTGYTY motifs? What about the number of CTCTGYTY motifs without REF6 binding? Also, it is beneficial to the general readers to provide more explanations on the purposes and results of the genomic analysis.

Response: We thank the reviewer for this suggestion to improve the manuscript. We have provided more details and explanations regarding the data analysis in the revised manuscript and the methods part. Within the overlapped 2,026 of 600 bp REF6 binding peaks from two biological replicates of REF6 ChIP-seq, 88% have one or more CTCTGYTY motifs. There are 31,911 CTCTGYTY motifs without REF6 binding within 24,786 regions defined by sliding a 600-bp window to contain motifs as many as possible.

Along the same line, my another concern is the biological significance of the anti-correlation between heterochromatic DNA methylation and REF6 binding. Since REF6 is a H3K27me3 demethylase and H3K27me3 is mainly in the body of euchromatic genes. It doesn't make too much sense for me that the authors draw strong conclusion based on the correlation from the DNA methylation in heterochromatin, unless the authors can demonstrate a distinct role of REF6 in heterochromatin independent of its H3K27me3 demethylase activity. Thus, it is critical to only focus the analysis on the biological relevant regions.

Response: Thank you for pointing out this. In our previous study, it is unclear why REF6 prefers to bind euchromatic regions and depletes from heterochromatic regions. REF6 specifically recognizes CTCTGYTY motifs in the genome, but only ~15% of such sequences in the genome are bound by REF6. We reasoned that some features of heterochromatic regions prevent REF6 from binding to such regions, such as DNA methylation, other epigenetic modifications, histone variants, and/or nucleosome positioning. Here we proved that CHG DNA methylation is at least one of the factors that prevents REF6 from binding to heterochromatin and helps REF6 to find its targets globally. It will be very interesting to dissect the reason why REF6 shall be repelled from heterochromatin as suggested by the reviewer in the future which we believe is out of the scope of the current manuscript.

The authors had previously generated H3K27me3 ChIP-seq in *ref6* mutants and generated a metaplot in Fig S1d. It is unclear what the purpose and how this metaplot is used in supporting their conclusion. It is important to correlate the hyper H3K27me3 regions in *ref6* with the CTCTGYTY motifs for further analysis.

Response: Thank you for the suggestion and we apologize for not describing this figure clearly. REF6 specifically recognizes CTCTGYTY motifs in the genome, and the chromatin features affect REF6 binding. The purpose of this metaplot (Fig S1d) is to explain why ~ 8.2% of REF6 targets are gene-body methylated (GbM) genes (Fig. S1b). We found that, even ~8.2% of REF6 targets are gene-body methylated (GbM) genes, REF6 binding sites are not overlapping with DNA methylation at these loci (REF6 binding sites are enriched in the TSS and TTS sites with low DNA methylation levels), which is consistent with our main conclusion. The reason of low H3K27me3 level in *ref6* in these regions may be REF6 is recruited to these regions by CTCTGYTY motifs (with original low H3K27me3 level), and functions to recruit other chromatin factors or transcription factors. Therefore, we removed the H3K27me3 data from the plot to avoid confusions to the readers.

Supplementary Figure 1. (d) Average profiles of REF6 ChIP-seq and DNA bisulfite sequencing signal at gene-body methylated (GbM) genes around the transcription start site (TSS) and transcription termination site (TTS).

Investigating the REF6 genome-wide occupancy in *ddcc* mutants is very nice (figure 4). The authors noted ectopic binding of REF6 to short TEs. But, why the number (22 out of 1220) is so low? Are 1220 the total REF6 ectopic binding peaks? Are they mostly in the heterochromatin? Of the 98% (~1200) REF6 ectopic binding peaks that don't overlap with TEs, what are they and what about their DNA methylation levels?

Response: We apologize for not describing clearly enough in the text. In the *ddcc* mutant, there are 1220 CHG hypomethylated differentially methylated regions (DMR) which contain CTCTGYTY-motifs and 22 of them were ectopic bound by REF6 as compared to that in Col. In the revised version, we performed replicate dataset 2 and identified 14 ectopic binding sites overlapping with previous dataset. In *ddcc* mutant, loss of non-CG methylation has minor effects on euchromatic structures associated with gene expression (Stroud, *et al.* 2014). Here we found these ectopic REF6 binding sites in *ddcc* are in short TE loci in euchromatic regions, and some of these ectopic binding events are associated with transcriptional activation of TEs or their neighboring protein-coding genes (Supplementary Fig. 6b).

Supplementary Figure 6. (b) Barplot of the RPKM value for AT2G17900 and AT2TE11570 from two replicates in Col and *ddcc* show significantly up-regulated gene expression level in *ddcc* (Stroud, *et al.* *Nat Struc Mol Biol.* 2014).

REF6 specifically recognizes CTCTGYTY motifs in the genome, but only ~15% of such sequences in *Arabidopsis* genome are bound by REF6,

suggesting some chromatin features prevent REF6 from potential binding sites, such as DNA methylation, other epigenetic modifications, histone variants, nucleosome positioning, higher-order chromatin structure, or cooperation with other transcription factors. Therefore, we conclude that DNA methylation is one of the factors that prevent REF6 from binding to CTCTGYTY motifs, and other factors are waiting for further exploration.

I noticed that the REF6 ChIP-seq signals appear to be general higher in *ddcc* compared to Col (e.g. *CUC1* control loci in Fig. 4b). The details regarding the REF6 ChIP-seq reads and replicates are missing. It is unclear whether the REF6 ChIP-seq has been biologically repeated, which is important as a certain variation in peak identification is expected between replicates. It is more critical to perform extensive normalization analysis for ChIP-seq data to normalize the overall signal to noise ratio between *ddcc* and Col samples. Although not required for this manuscript, this reviewer highly recommended normalizing across samples using spiked-in methods.

Response: We thank the reviewer for this great suggestion. To prove the reproducibility of identified ectopic binding sites, we performed another biological replicate of REF6 ChIP-seq which allows us to distinguish the real ectopic binding events from the potential noise. Overall, two batches of ChIP-seq data showed high Pearson correlation coefficient between each other (Supplementary Fig. 5). Although the number of ectopic binding peaks of REF6 is relatively low in the *ddcc* mutant, the binding affinity to these sites are significantly and reproducibly high in *ddcc* mutant (Fig. 5b and Supplementary Fig. 6a), indicating these ectopic binding peaks are *bona fide* REF6 binding sites in *ddcc* mutant.

Detailed high-throughput sequencing data is shown in Supplementary Table 1 (see below). With regard to the normalization analysis for the ChIP-seq data, besides normalizing with the sequencing library size, we selected the

overlapping peaks from the two replicates to calculate binding intensity (reads per million, RPM) by subtracting control signal from treatment signal. In addition, ChIP-qPCR results of several ectopic binding sites also support the ChIP-seq results. The spiked-in strategy is a powerful method in normalizing across samples, we will implement this method in our future work.

Supplementary Figure 5: Scatterplots of the normalized ChIP-seq signal intensity in \log_2 scale over merged peak regions of two replicates show high correlation in replicates.

Figure 5. (b) Genome-browser view of REF6 binding and DNA methylation in Col and the *ddcc* at the *AT2TE32120* and *AT5TE57090* loci. The *CUC1* locus was used as the control.

Supplementary Figure 5. (a) Genome-browser view of REF6 binding and DNA methylation in Col and the *ddcc* at the AT1G31210, AT2TE11570, and AT2TE38575 loci.

As stated above that REF6 is a H3K27me3 demethylase and H3K27me3 is mainly in the gene body regions, mostly with CG methylation. In Fig.1b, the CG DNA methylation level had a most obvious change among the three DNA methylation contexts. Thus, it makes more sense to test the role of CG methylation in REF6 binding (e.g REF6 ChIP-seq in *met1* or *ddm1* mutants). The authors may also consider (not required) perform REF6 ChIP-seq in *imb1* or *ros1dme1dme2* triple mutants (hypermethylation mutants) to provide more evidence.

Response: This is a great idea! To explore the role of CG methylation in REF6 targeting, we performed REF6 ChIP-seq in *met1-1* mutant, which loss CpG methylation in gene body (Catoni *et al*, *EMOB J.*, 2017). However, we found no obvious changes in REF6 binding between *met1* and Col (Supplementary Figure 1.1 below). Therefore, we concluded that gene body CG methylation does not affect REF6 binding. Performing REF6 ChIP-seq

in *ibm1* or *ros1dme1dme2* triple mutants are great suggestions. However, we found the hyper-CHG DMR sites in *ibm1* and *rdd* mutants show very little overlap with REF6-binding sites in wild-type plants, suggesting REF6 and these DNA/histone demethylases regulate different sets of genes (Supplementary Figure 1.2 below).

Supplementary Figure 1.1. (a) High correlation between two replicates of REF6 ChIP-seq in *met1*. (b) REF6 ChIP-seq signal (measured by mapped reads per million, RPM) at the overlapping regions of calling peaks from two biological replicates in wide-type Col and *met1* mutant. The red dots represent the statistically significantly differential binding intensity, while those on the left of the dotted line are ectopic binding sites.

Supplementary Figure 1.2. Gain-of-CHG methylation in *ibm1* and *rdd* mutants block REF6-binding to target genes.

The manuscript is well written in general, but may benefit to have more details in the results as well as the method sections and more discussions. It is also important to include a table describing the genomic sequencing data.

Response: Thank you for the suggestions to help improving our manuscript. We have provided the information in our modified manuscript and the supplementary table describing the genomic sequencing data.

Supplementary Table 1: Summary of ChIP-Bisulfite-sequencing data analysis.

Library	Library Type	Total reads	Clean reads	BS-seeker unique-hits reads (% of total reads)
ref6 ChBS-Input-rep 1	ChIP-BS	41,210,672	41,204,760	14,772,837 (35.85%)
Col REF6-ChBS-rep 1	ChIP-BS	39,368,136	39,359,562	10,366,454 (26.34%)
ddcc REF6-ChBS-rep1	ChIP-BS	38,909,862	38,892,547	11,471,341 (29.50%)
ref6 ChBS-Input-rep 2	ChIP-BS	42,465,152	42,453,991	17,662,835 (41.60%)
Col REF6-ChBS-rep 2	ChIP-BS	39,444,734	39,436,424	13,961,957 (35.40%)
ddcc REF6-ChBS-rep 2	ChIP-BS	38,161,444	38,155,586	9,783,749 (25.64%)
Library	Library Type	Total reads	Total mapped reads (% of total)	Unique mapped reads (% of total mapped)
ref6 -Input-rep 1	ChIP-seq	5,718,280	2,490,406 (43.55%)	1,733,935 (69.62%)
Col REF6-IP-rep 1	ChIP-seq	3,754,404	2,251,278 (59.96%)	1,724,947 (76.62%)
ddcc REF6-IP-rep 1	ChIP-seq	5,199,673	2,855,213 (54.91%)	2,199,248 (77.02%)
ref6 -Input-rep 2	ChIP-seq	22,145,136	9,986,503 (45.10%)	7,053,860 (70.63%)
Col REF6-IP-rep 2	ChIP-seq	22,904,799	11,978,451 (52.30%)	8,717,474 (72.78%)
ddcc REF6-IP-rep 2	ChIP-seq	22,686,457	12,127,221 (53.46%)	9,068,274 (74.78%)
met1 REF6-IP-rep 2	ChIP-seq	21,122,919	7,947,180 (37.62%)	5,778,421 (72.71%)
ref6 -Input-rep 3	ChIP-seq	26,190,901	11,309,686 (43.18%)	7,711,563 (68.19%)
met1 REF6-IP-rep 1	ChIP-seq	25,446,803	13,537,985 (53.20%)	9,271,120 (68.48%)

L156, change “significant” to “great” as no significant test has been done in the ITC experiments.

Response: Thank you for pointing out this mistake. We have changed “significant” to “great”.

Reviewer #3 (Remarks to the Author)

In the current submission “DNA methylation repels REF6 targeting in Arabidopsis” Qiu et al. demonstrate that the Zinc Finger domain of this H3K27 demethylase is sensitive to DNA methylation both in vitro and in vivo and reveal the structural basis for this sensitivity. Furthermore, they demonstrate that by altering the DNA methylation landscape in the *ddcc* mutant background, REF6 localizes to new target sites. While there is a small, but growing number of TFs shown to be sensitive to DNA methylation based on in vitro analyses, there remains little evidence regarding the effects of such sensitivities in vivo. The data by Qiu et al represent an excellent in vivo example in which DNA methylation affects chromatin association and provide what may be the first example wherein a methylation sensitive DNA binding module connects two epigenetics modifications (DNA methylation and H3K27 methylation). The findings of this manuscript are well supported by the data and I only have minor comments (see below).

minor comments:

line 63, the sentence about the role of DNA methylation in gene silencing is redundant with the second sentence of the same paragraph and could be removed.

Response: Thank you for pointing out this redundancy. We have removed this sentence.

As part of the last paragraph of the introduction the authors should also mention the recent work from the Ecker lab in looking at the methylation sensitivity of

many TFs via DAPseq assays.

Response: Thank you for pointing out this reference. We have added this recent work from Ecker lab to the manuscript: A high-throughput TF binding site discovery method, namely DNA affinity purification sequencing (DAP-seq), allows to identify the potentially genomic binding sites of several hundreds of TFs (Bartlett, et al. 2017). Using this method, they found that 76% of Arabidopsis TFs they studied were sensitive to DNA methylation (O'Malley, et al. 2016).

The phrase “intrinsic DNA methylation unfavorable DNA binding activity” is quite awkward. Perhaps “methylation sensitive DNA binding activity” would be better?

Response: Thank you for the wording. We have changed it into “methylation sensitive DNA binding activity”.

The authors use two different nomenclatures to refer to methylated cytosines (mC and 5mC). As these are the same thing, the authors should stick to a single naming system to avoid unnecessary confusion.

Response: Thank you for pointing out this mistake. We have changed all mC into 5mC.

In Figure 3e, I believe that the C5 and G5 labels are switched.

Response: Thank you for pointing out this mistake. We have corrected it.

line 164-65 the names of DRM2 and CMT3 were already defined in the introduction and thus do not need to be redefined here.

Response: Thank you for pointing out this duplication. We have removed the redefinition of DRM2 and CMT3 in line 164-165.

Information regarding the locations of the REF6 peaks from the ChIP experiments (wt, *ddcc*, etc) should be provided as well as general information about the ChIP libraries (coverage, mapping etc).

Response: Thank you for the suggestion. We have provided the general information about the ChIP libraries in Supplementary Table 1, and the REF6 ectopic peaks information in Supplementary Table 4.

Supplementary Table 1: Summary of ChIP-Bisulfite-sequencing data analysis.

Library	Library Type	Total reads	Clean reads	BS-seeker unique-hits reads (% of total reads)
ref6 ChBS-Input-rep 1	ChIP-BS	41,210,672	41,204,760	14,772,837 (35.85%)
Col REF6-ChBS-rep 1	ChIP-BS	39,368,136	39,359,562	10,366,454 (26.34%)
ddcc REF6-ChBS-rep1	ChIP-BS	38,909,862	38,892,547	11,471,341 (29.50%)
ref6 ChBS-Input-rep 2	ChIP-BS	42,465,152	42,453,991	17,662,835 (41.60%)
Col REF6-ChBS-rep 2	ChIP-BS	39,444,734	39,436,424	13,961,957 (35.40%)
ddcc REF6-ChBS-rep 2	ChIP-BS	38,161,444	38,155,586	9,783,749 (25.64%)
Library	Library Type	Total reads	Total mapped reads (% of total)	Unique mapped reads (% of total mapped)
ref6 -Input-rep 1	ChIP-seq	5,718,280	2,490,406 (43.55%)	1,733,935 (69.62%)
Col REF6-IP-rep 1	ChIP-seq	3,754,404	2,251,278 (59.96%)	1,724,947 (76.62%)
ddcc REF6-IP-rep 1	ChIP-seq	5,199,673	2,855,213 (54.91%)	2,199,248 (77.02%)
ref6 -Input-rep 2	ChIP-seq	22,145,136	9,986,503 (45.10%)	7,053,860 (70.63%)
Col REF6-IP-rep 2	ChIP-seq	22,904,799	11,978,451 (52.30%)	8,717,474 (72.78%)
ddcc REF6-IP-rep 2	ChIP-seq	22,686,457	12,127,221 (53.46%)	9,068,274 (74.78%)
met1 REF6-IP-rep 2	ChIP-seq	21,122,919	7,947,180 (37.62%)	5778421 (72.71%)
ref6 -Input-rep 3	ChIP-seq	26,190,901	11,309,686 (43.18%)	7,711,563 (68.19%)
met1 REF6-IP-rep 1	ChIP-seq	25,446,803	13,537,985 (53.20%)	9,271,120 (68.48%)

Supplementary Table 4: Ectopic binding sites of REF6 in *ddcc*.

Chr	Peak_start	Peak_end	Gene_ID	Type	Gene_annotation
Chr1	11147736	11148077	AT1TE36030	transposable_element	ATCOPIA25
Chr1	11152736	11152894	AT1TE36040	transposable_element	ATCOPIA52
Chr1	16238193	16238503	AT1G43145	protein_coding	unkown protein
Chr1	22795991	22796210	AT1G61732	protein_coding	encodes a microRNA
Chr1	17680960	17681313	AT1TE58705	transposable_element	LTR/copia
Chr2	1468879	1469430	NA	intergenic	
Chr2	2578356	2578725	AT2TE11570	transposable_element	ATCOPIA50
Chr2	2580213	2580887	AT2G06500	protein_coding	hAT family dimerisation domain
Chr2	9122126	9122682	AT2TE38575	transposable_element	ATCOPIA74
Chr2	7776685	7777792	AT2TE32120	transposable_element	LINE/L1
Chr3	14861102	14861453	AT3TE60635	transposable_element	ATCOPIA34
Chr4	7060763	7061163	AT4TE30625	transposable_element	ATLINEIII
Chr5	1683301	1683690	AT5G05630	protein_coding	Polyamine uptake transporter 3
Chr5	15813699	15814483	AT5TE57090	transposable_element	ATHATN2

If available additional information regarding the effects of the new REF6 targets sites in the *ddcc* background, like H3K27me levels or gene expression changes, would make an excellent addition to this manuscript.

Response: Thank you for the suggestion. In *ddcc* mutant, loss of non-CG methylation has minor effects on euchromatic structures associated with gene expression (Stroud, *et al.* 2014). Here we found that these ectopic REF6 binding sites in *ddcc* are in short TE loci in euchromatic regions, and some of these ectopic binding events are associated with transcriptional activation of TEs or their neighbor protein-coding genes (Supplementary Fig. 6). It implies that REF6 binding may underlie the activation of gene transcription by changing chromatin states.

Supplementary Figure 6. (b) Barplot of the RPKM value for AT2G17900 and AT2TE11570 from two replicates in Col and *ddcc* show significantly up-regulated gene expression level in *ddcc*.

In figure 4a, the legend refers to “the thick red dot” but there are many red dots.

Response: Thank you for pointing out this mistake. We have changed it into “the thick red dots”.

In figure 4d, the legend refers to chromosome 1 as “chromosome I”.

Response: Thank you for pointing out this mistake. We have changed it as “chromosome 1” in Figure 5d, and “chromosome 2, 3, 4 and 5” in Supplementary Figure 7.

Reviewer #4 (Remarks to the Author):

In this paper, Qiu et al report an “antagonistic mechanism” between REF6 and DNA methylation. This interesting finding is supported by structural and seq-based genomic profiling studies. Overall, this story is conceptually important and the regulatory mechanism is interesting. I would like to recommend publication of this manuscript if the following concerns are properly addressed.

Major point:

1) According to ITC assays (Figure 3f), the DNA methylation only results in

about twofold reduction of the binding affinity. REF6 binds to methylated DNA strands at $K_D=77-173$ nM, which is still a very strong binding event. In this case, I do not hold the view that “5mCs lead to significant reduction in the binding affinity to the CTCTGYTY motif” in vitro. According to the EMSA data (Figure 2), it is hypermethylation of cytosine but not single site 5mC that functions to repel REF6. Therefore, the authors should carefully revisit their conclusion and squarely conclude their observations. An ITC titration using hypermethylated DNA substrate should be performed. If proven true, it's better to change the current title to “DNA hypermethylation repels REF6 targeting in Arabidopsis”.

Response: We thank the reviewer for this great suggestion. According to the EMSA result, we found 5mC at different sites show different effects on binding affinity. 5mC at C₅ site severely reduce the binding affinity, while 5mC at C₁ or C₃ alone doesn't have such big effects. As Reviewer #4 mentioned, the DNA methylation only results in about twofold reduction of the binding affinity in previous ITC results. We reasoned the difference between two assays may be caused by difference in the binding buffers. We adjusted the concentration of NaCl in ITC buffer from 300 mM to 150 mM, and performed the ITC assay with newly purified REF6-ZnF and newly synthesized DNA probes. Consistent with the previous results, 5mC₁ and 5mC₃ do not affect REF6-ZnF binding to DNA probes. However, 5mC₅ completely abolished the protein-DNA interaction (Fig. 4), which is consistent with our EMSA result. We repeated this experiment several times and got consistent result among replicates. In view of this result, we further performed ITC assay with 5mC₁+5mC₃ probe and found that 5mC₁+5mC₃ completely abolished the protein–DNA interaction, which is also consistent with our EMSA result. Therefore, we replaced the old data with these new data in the revised manuscript.

Figure 4. (f) ITC assays showing decreased interaction between REF6-ZnFs and methylated DNA probes. NDB, no detectable binding.

Similarly, the hypermethylation state of the “CTCTGYTY” motif should be examined and confirmed *in vivo*. This is can be explored by single-base-resolution sequencing analysis of the genome DNA.

Response: Thank you for the suggestion. To further confirm that REF6 preferentially bind to unmethylated DNA motifs *in vivo*, we performed REF6 ChIP-bisulfite-sequencing (ChIP-BS-seq) (Statham, *et al. Genome Research. 2012*) in Col, compared with *ddcc*, in which non-CG methylation is completely lost (Supplementary Table 1). The results showed an anti-correlated profile between REF6 binding signal and DNA methylation level at REF6 binding peaks (Fig. 2a). DNA methylation level of REF6 binding peaks identified by ChIP-BS-seq in Col are as low as that in *ddcc*, indicating that there is not significant difference between wild-type Col and *ddcc* for differential sensitivity to non-CG DNA methylation (Fig. 2b). Moreover, REF6 bound DNA showed lower methylation level compared to that in WGBS data, indicating that REF6 bound DNA was depleted for DNA methylation while the methylation at REF6 binding sites seen in WGBS data may come from DNA without REF6 binding in some cell types (Fig. 2b). These results give direct evidence supporting that REF6 prefers to bind hypomethylated DNA in the *Arabidopsis* genome. As to the CHG hypermethylation in CTCTGYTY motif, no REF6 binding is found.

Figure 2. ChIP-BS-seq shows REF6 prefers to bind hypomethylated regions in *Arabidopsis* genome.

(a) Heat maps of REF6 occupancy and DNA methylation level in 1.0 Kb surrounding REF6 binding peaks.

(b) Box plots showing average non-CG DNA methylation level of *in vivo* REF6 binding regions. The DNA methylation levels of REF6 binding regions in Col and *ddcc* were measured by ChIP-BS-seq data, while those in REF6-bound (REF6+) and REF6-unbound (REF6-) regions were measured by whole genome bisulfate sequencing in Col from public data. *In vivo* REF6 binding regions and REF6+ show significant difference (Mann-Whitney U test, $P < 2.2e-16$) to REF- motifs. NS, not significant.

2) Additional perturbation studies should be performed. For example, based on the structural analysis, the authors should be able to design REF6 mutants that can tolerate 5mC. It would be nice to investigate the functional impact if the mutant REF6 is introduced in plant. Such an effort will significantly improve the quality of the story.

Response: This is a great suggestion. We expressed REF6-ZnF with a Trp1311 to Ala mutation (W1311A) and performed EMSA and ITC with *NAC004* probe. We found that W1311A mutation abolished the interaction between REF6-ZnF and DNA probe, indicating Trp1311 is important for

REF6-ZnF binding to DNA. Therefore, design of such a mutation version of ZnF is still challenging even we know W1311 is the methylation-sensitive site. But this is a promising direction that will improve our understanding for the function of REF6 in the future.

Figure. W1311 is essential for REF6-ZnF binding to DNA. EMSA and ITC assay showed W1311A can not bind to *NAC004*.

Minor points:

1) A complete ITC fitting parameters should be provided.

Response: Thank you for pointing out this. A complete ITC fitting parameters have been provided in Supplementary Table 3.

Supplementary Table 3. ITC fitting parameters.

DNA probe	DNA sequence	N	K	Kd (nM)
NAC004	5'-TTCTCTGTTTTG-3' 3'-AAGAGACAAAAC-5'	1.17±0.00423	1.36E7±1.42E6	73.5±7
NAC004_5mC ₁	5'-TTC(m)TCTGTTTTG-3' 3'-AAGAGACAAAAC-5'	1.06±0.00554	1.52E7±2.40E6	65.7±4.9
NAC004_5mC ₃	5'-TTCTC(m)TGTTTTG-3' 3'-AAGAGACAAAAC-5'	1.03±0.0132	9.06E6±2.64E6	110.3±3.8
NAC004_5mC ₁ +5mC ₃	5'TTC(m)TC(m)TGTTTTG-3'	No detectable binding		

	3'- AAGAGACAAAAC -5'	
NAC004_5mC ₅	5'-TTCTCTGTTTTG-3' 3'AAGAGAC(m)AAAAC-5'	No detectable binding

2) Figure 3e, C5 and G5' are mislabeled. In addition, a role of W1311 should be confirmed by mutagenesis studies.

Response: Thank you for pointing out this. We have changed the Figure 3e correctly. We expressed REF6-ZnF with a Trp1311 to Ala mutation (W1311A) and performed EMSA and ITC with NAC004 probe. We found that W1311A mutation abolished the interaction between REF6-ZnF and DNA probe, indicating Trp1311 is not only important for preventing REF6-ZnF from binding to methylated DNA, but also essential for REF6-ZnF binding to DNA.

Figure. W1311 is essential for REF6-ZnF binding to DNA. EMSA and ITC assay showed W1311A can not bind to NAC004.

3) The proposed mechanistic module in Figure 5 is not supported by the present data. In fact, the binding affinity of REF6 to methylated CTCTGYTY motif is not weak, and even corresponding complex crystal structures have even been determined. The red blocking arrow does not properly reflect the experimental observations.

Response: Thank you for pointing out this. According to our EMSA and new ITC results, $5mC_1+5mC_3$ and $5mC_5$ completely abolished the protein–DNA interaction. Therefore, we thought that the proposed mechanistic model can reflect the experimental observations.

4) In supplementary Table 1, the unit cell angles should be fixed to integral numbers, e.g. (90, 90, 90). These values are not measured ones for the current space groups. Also please double check the space group of ZnF2-4-NAC004-5mC1. The unit cell angles of (90, 90, 120) is not consistent with the P4(3) space group.

Response: Thank you for pointing out this. In Supplementary Table 1, we have changed the unit cell angles to integral numbers, and changed the space group of ZnF2-4-NAC004-5mC₁ to P3₁.

REVIEWERS' COMMENTS:

Reviewer #1 (Remarks to the Author):

The authors have performed all of the appropriate experiments and they are of high quality. They show that DNA methylation to a certain extent can repel REF6 in vitro. The in vivo data, however, does not support this claim. It does show that REF6 doesn't bind to methylated regions, but as there are only 14 ectopic REF6 binding sites indicate that this is not a major function of REF6 activity. Considering there are thousands of REF6 binding sites in the genome this is especially hard to reconcile with the proposed functions. The binding of REF6 to unmethylated regions is more likely due to indirect effects such as DNA methylation and H3K27me3 not being colocalized in the Arabidopsis genome. I fail to see the major biological advance of this study even though I don't doubt any of the experimentation.

Reviewer #2 (Remarks to the Author):

In this revision, the authors have addressed most of my concerns and comments by performing additional experiments and extensively edited the manuscript. Thus, it is now suitable for publication in the Nature Communications.

Xuehua Zhong

Reviewer #4 (Remarks to the Author):

Most of my concerns have been properly addressed with the new titration and sequencing data. It is a pity that the authors failed to identify a methylation-tolerant mutant of REF6 for perturbation studies. Hopefully this will be achieved in future efforts. Recommend for publication!

In Supplementary Table 3, the fitting parameters of ΔH and $T\Delta S$ should also be included.

We thank all the reviewers for their constructive suggestions and have revised our manuscript accordingly. A point-to-point response is below with our responses highlighted in **blue**.

REVIEWERS' COMMENTS:

Reviewer #1 (Remarks to the Author):

The authors have performed all of the appropriate experiments and they are of high quality. They show that DNA methylation to a certain extent can repel REF6 *in vitro*. The *in vivo* data, however, does not support this claim. It does show that REF6 doesn't bind to methylated regions, but as there are only 14 ectopic REF6 binding sites indicate that this is not a major function of REF6 activity. Considering there are thousands of REF6 binding sites in the genome this is especially hard to reconcile with the proposed functions. The binding of REF6 to unmethylated regions is more likely due to indirect effects such as DNA methylation and H3K27me3 not being colocalized in the Arabidopsis genome. I fail to see the major biological advance of this study even though I don't doubt any of the experimentation.

Response: We appreciate your comments and edit the discussion part according to editor's suggestion. In our opinion, REF6 ectopic binding sites in *ddcc* mutant likely result from not only loss of DNA methylation, but also changes of other chromatin features. However, we believe that loss of DNA methylation is indispensable for REF6 targeting to these loci. As shown in Fig.3 and Fig.4f, DNA methylation in CTCTGYTY-motif can dramatically reduce ZnF-binding affinity *in vitro*, so loss of DNA methylation is necessary but not sufficient for ectopic REF6-binding to specific loci *in vivo*.

Besides DNA methylation and H3K9me2, other epigenetic markers, such as H2A.W, are also important for heterochromatin maintenance (Yelagandula et al, Cell, 2014). It's

possible that mutations in one pathway are not sufficient to “unlock” the heterochromatic structure. It will be interesting to test whether REF6 binds to ectopic heterochromatin regions in higher order mutants.

Reviewer #2 (Remarks to the Author):

In this revision, the authors have addressed most of my concerns and comments by performing additional experiments and extensively edited the manuscript. Thus, it is now suitable for publication in the Nature Communications.

Xuehua Zhong

Response: We appreciate your suggestions to improve our manuscript.

Reviewer #4 (Remarks to the Author):

Most of my concerns have been properly addressed with the new titration and sequencing data. It is a pity that the authors failed to identify a methylation-tolerant mutant of REF6 for perturbation studies. Hopefully this will be achieved in future efforts. Recommend for publication!

In Supplementary Table 3, the fitting parameters of ΔH and $T\Delta S$ should also be included.

Response: We appreciate your suggestions to improve our manuscript. We hope a methylation-tolerant mutant of REF6 can be identified in the future. We have added the fitting parameters of ΔH and $T\Delta S$ in revised manuscript.

Supplementary Table 3. ITC fitting parameters.

DNA probe	DNA sequence	N	K	Kd (nM)	ΔH cal/mol	T ΔS cal/mol
NAC004	5'-ttctctgtttg-3' 3'-aagagacaaaac-5'	1.17±0.00423	1.36E7±1.42E6	73.5±7	-6145±34.82	300
NAC004_5mC ₁	5'-ttc(5mC)tctgtttg-3' 3'-aagagacaaaac-5'	1.06±0.00554	1.52E7±2.40E6	65.7±4.9	-2856±25.33	582.5
NAC004_5mC ₃	5'-ttctc(5mC)tggtttg-3' 3'-aagagacaaaac-5'	1.03±0.0132	9.06E6±2.64E6	110.3±3.8	-2162±43.59	615
NAC004_5mC ₁ + 5mC ₃	5'-ttc(5mC)tc(5mC)tggtttg-3' 3'-aagagacaaaac-5'	N.D.				
NAC004_5mC ₅	5'-ttctctgtttg-3' 3'-aagagac(5mC)aaaac-5'	N.D.